# Low-Grade Adenosquamous Carcinoma of the Breast: A Single-Center Retrospective Study and a Systematic Literature Review

**DOI:** 10.3390/cancers16244246

**Published:** 2024-12-20

**Authors:** Anselm Tamminen, Pia Boström

**Affiliations:** 1Department of Plastic and General Surgery, Turku University Hospital, 20520 Turku, Finland; 2Department of Surgery, University of Turku, 20014 Turku, Finland; 3Department of Pathology, Turku University Hospital, 20520 Turku, Finland; pia.bostrom@varha.fi; 4Department of Pathology, University of Turku, 20014 Turku, Finland

**Keywords:** breast cancer, low-grade adenosquamous carcinoma, metaplastic carcinoma

## Abstract

Low-grade adenosquamous carcinoma (LGASC) is a rare and indolent subtype of metaplastic breast carcinoma (MpBC), comprising less than 0.05% of breast cancer cases. Unlike other MpBCs, LGASC is associated with an excellent prognosis and minimal risk of metastasis. It often presents as a small palpable periareolar mass, with its diagnosis challenging in imaging and histopathological studies. LGASC is frequently misdiagnosed due to its overlap with benign sclerosing lesions, particularly in core needle biopsies. The optimal treatment involves breast-conserving surgery with negative margins. Omitting sentinel lymph node biopsy appears to be feasible due to the very low metastatic risk. Adjuvant therapies like chemotherapy, radiation, or hormonal therapy are generally unnecessary. Based on the literature and a single-center review of three cases, LGASC should be recognized as a distinct entity to avoid overtreatment and ensure appropriate, conservative management.

## 1. Introduction

Metaplastic breast carcinoma (MpBC) is a rare type of breast cancer, accounting for less than 1% of all breast cancer cases. It is often presented as a cohesive entity, but it actually consists of six unrelated subtypes: spindle cell carcinoma, squamous cell carcinoma, metaplastic carcinoma with heterogeneous mesenchymal differentiation, low-grade adenosquamous carcinoma, low-grade fibromatosis-like metaplastic carcinoma, and mixed metaplastic carcinoma. Each of these subtypes has specific characteristics with certain similarities to each other, but also with important differences. Commonly, their tumor cells differentiate into squamous or mesenchymal-looking elements [1].

Low-grade adenosquamous carcinoma (LGASC) is one of the rarest subtypes of MpBC. It was recently estimated that only 155 cases have been reported in the literature [2]. Based on the data from large breast cancer databases, it has been estimated that one out of 4000 to 5000 breast cancer cases are LGASC [3,4].

MpBCs are, predominantly, a triple-negative breast cancer (TNBC), with the poorest overall prognosis among all breast cancer types independent of the receptor status [5,6]. Although most variants of MpBC are indeed aggressive and chemoresistant, in contrasr, LGASC is an indolent type of breast cancer with an excellent prognosis [7]. Therefore, it is particularly important to distinguish LGASC cases from the ‘entity’ called MpBC [8].

Due to the rarity of LGASC, clinicians may detect this disease only once or twice during their career. Additionally, most scientific evidence of LGASC is based on case reports or series, rather than on systematic research [4,6]. As a consequence, the demand for individualized treatment for LGASC may go unnoticed, and even if recognized, the lack of coherent data and treatment guidelines could lead to mistreatment [6,9]. Since MpBC is typically aggressive, patients with MpBC are usually treated more intensively, undergoing more extensive breast surgery, more frequent ALND, and more often receiving chemotherapy compared to patients with other subtypes of breast cancer [6].

LGASC is considered an indolent type of breast cancer, in contrast to the typically aggressive nature of other subtypes of MpBCs. Accordingly, it should be treated more conservatively than the more aggressive types of breast cancer. However, clear treatment guidelines are lacking. NCCN breast cancer guidelines state that there are limited available data on the treatment of LGASC, but that it has a favorable prognosis even without systemic adjuvant therapy. The guideline does not address LGASC as a separate entity but mentions it alongside other breast cancer types with a favorable prognosis, stating that systemic treatments may not be necessary for localized disease, although the data on this are limited, and the need for radiation therapy is not addressed [10]. 

To expand knowledge on this topic, we conducted a single-center retrospective analysis of LGASC patients treated between 2010 and 2022, along with a systematic literature review to summarize our current understanding of this rare entity known as LGASC.

## 2. Materials and Methods

### 2.1. Patient Data from Single Tertiary University Hospital

Patient data from breast cancer patients treated in our tertiary university hospitals between the years 2010 and 2022 were acquired from Auria Clinical Informatics Center. The data were manually evaluated and the data of patients with LGASC were collected, and correctness was ensured using electronic health records.

### 2.2. Search Strategy for Systematic Literature Review

The systematic review was conducted in accordance with the PRISMA (Preferred Reporting Items for Systematic Reviews and Meta-Analyses) guidelines, where applicable, considering the nature of the literature, which consists mainly of case reports and short case series. The Embase, Medline, and PubMed databases were searched in September 2024 using the terms ‘breast’ or ‘mammary’ and ‘adenosquamous carcinoma’. Only articles published in English were considered. The titles were initially screened, and those that did not appear to address LGASC of the breast were excluded. The remaining abstracts were reviewed, and potentially relevant articles were retrieved in full text. The reference lists of the retrieved articles were cross-checked to ensure that all relevant studies were included in the analysis. The review was not registered.

The following information was collected: the period in which the patient information was collected, the number of patients, their age, the presentation and size of the tumor, the surgery performed, information about local or distant metastasis at the time of diagnosis, adjuvant treatments and follow-up information. As a result, a synthesis and descriptive analysis of the published cases are presented.

## 3. Results

### 3.1. Low-Grade Adenosquamous Carcinoma Patient Cases in Single Center Data

In total, 6462 patients were diagnosed with breast cancer between the years 2010 and 2022. Three of the patients (0.05%, 95% CI, 0.01–0.014%) were diagnosed with LGASC. The data of the patients are presented in Table 1 (clinical presentation), Table 2 (histopathological data) and Table 3 (treatment and surveillance data). Images of the histopathological samples are provided as Appendix A.

### 3.2. Results of the Systematic Literature Review

After electronic and manual de-duplication, the exclusion of irrelevant articles, and manual cross-referencing, 62 articles were retrieved and read in full, with 35 papers reporting patient data (Figure 1).

Details of the published patient cases [11,12,13,14,15,16,17,18,19,20,21,22,23,24,25,26,27,28,29,30,31,32,33,34,35,36,37,38,39,40,41,42,43,44,45] are presented in full in Appendix A.

#### 3.2.1. Overview of the Data

In the detailed analysis, it was found that the patients included in one article (*n* = 3) were also part of the data in another article with common authors. Consequently, the data from the former article were excluded from the analysis. The remaining 34 articles reporting patient data included a total of 227 patients. The studies exhibited significant heterogeneity in their patient sample sizes, with 59% (20 of 34) reporting only a single case. The three oldest studies accounted for a total of 62 patients, representing more than a quarter of all reported cases. These studies also shared, at least in part, the same authors, making it likely that data from the same patients were included in multiple studies.

Publications with multiple patients typically reported study periods spanning 10–30 years, highlighting the rarity of LGASC. One of the most recent patient series included 34 patients, but the data were available only as a conference abstract, which nevertheless provides the essential information [42]. In one study, a high-grade adenosquamous carcinoma was included in the data, but the information for LGASC patients was not reported separately, complicating the analysis [40]. Overall, the data presented in the articles are inconsistent, reflecting the fact that the diagnosis and treatment of breast cancer have changed substantially in recent decades, as well as indicating that the literature originates from physicians across various specialties and areas of interest.

#### 3.2.2. Pathophysiology of LGASC

The pathophysiology of LGASC of the breast remains largely speculative due to its rarity and the limited number of studies addressing this topic. Evidence suggests that LGASC is very often associated with benign lesions, such as fibrocystic changes, radial scars, and papillomas [12,33]. Therefore, it is hypothesized that adenosquamous proliferation linked to these benign lesions may serve as a precursor to LGASC [46].

#### 3.2.3. Clinical Presentation

Although most studies state that LGASC is often diagnosed in postmenopausal women aged 50 years or older, the patient data only partially support this claim, as approximately 30% of the patients were under 50 years of age at the time of diagnosis. The youngest reported patient was just 19 years old [7,22]. Notably, a single case of male patient with LGASC was also presented [47]. Despite being a very rare form of breast cancer, bilateral LGASC has been reported in at least three patients [29].

Patients predominantly present with a palpable mass (80%, 154/192). Only 30 patients (16%) were diagnosed after mammography, typically as part of a screening program. This may be somewhat unexpected, as many patients belong to age groups eligible for screening mammography, and LGASC appears to be a slowly progressing type of breast cancer.

There appears to be a strong tendency for the tumor to occur in the periareolar region. Associated nipple symptoms, such as nipple retraction and hardening, and rarely nipple discharge, may also lead to an LGASC diagnosis [19,27]. However, several studies do not specify the exact location of LGASC within the breast [26].

The size of the tumor is rather consistently less than 25 mm at the time of diagnosis (mean of overall data: 19 mm). In Kawaguchi and Shin’s data on 30 patients, 29 presented with a palpable lump, and tumor size information was available for 19 patients; only one had a tumor larger than 25 mm. One patient had an LGASC tumor measuring 50 mm, but her LGASC was initially misdiagnosed as syringomatous adenoma, with the correct diagnosis made five years after the initial diagnosis [24].

Only one of the 171 cases (0.6%) was reported to have lymph node metastasis at presentation. This patient, reported by van Hoeven et al. in 1993, also had initial distant metastasis [12]. Additionally, one patient in Cartagena et al.’s data presented with isolated tumor cells but had concurrent ipsilateral ductal carcinoma in situ (DCIS) [42].

#### 3.2.4. Imaging Studies

MpBCs are reported to have a more benign appearance on mammographic and sonographic imaging, despite their typically aggressive nature. In contrast, LGASC appears to exhibit more classically suspicious imaging features, despite being an indolent type of breast cancer. LGASC may share characteristics with both malignant and benign neoplasms, and so far, no distinctive mammographic or ultrasound findings have been reported [26,40].

#### 3.2.5. Mammography

Mammographically, most LGASC cases present with architectural distortion and suspicious irregular masses, often with spiculated or indistinct margins. The architectural distortion may be occult on conventional mammography but is more frequently detected with tomosynthesis. High-density masses and asymmetry are reported infrequently. Microcalcifications are uncommon [44], and the tumor only rarely presents as an oval, circumscribed mass, unlike the majority of metaplastic cancers.

However, LGASC may present with concurrent sclerosing lesions and associated calcifications, which may eventually lead to its detection. In such cases, the LGASC tumor is usually significantly smaller than the microcalcifications initially identified [26,31,40,48]. In a few reported cases, the LGASC lesion is not visible on mammograms, particularly in instances of heterogeneous or very dense breast tissue. In such cases, the lesion is usually detected on ultrasound [48].

#### 3.2.6. Ultrasound

In ultrasound studies, LGASC typically presents as a suspicious irregular hypoechoic mass. A clearly circumscribed mass is unusual but occasionally encountered [32]. Margins may be indistinct or microlobulated. In some cases, axillary lymph nodes may show cortical thickening, but axillary lymph node metastases are rare [26,40].

#### 3.2.7. Magnetic Resonance Imaging

Very few published studies describe the appearance of LGASC on magnetic resonance imaging (MRI). A mass with spiculated margins, heterogeneous enhancement, and early washout has been identified as the most common abnormality associated with LGASC. MpBC typically presents as an irregular, spiculated mass on MRI, and the few reported cases of LGASC undergoing preoperative MRI appear to share features commonly reported for MpBC. Dynamic contrast-enhanced MRI likely does not provide additional information beyond regular MRI and is unlikely to help differentiate LGASC from other lesions. However, due to the limited number of reported cases, more data are needed to validate these assumptions. In the reported cases, no axillary or internal mammary lymphadenopathy has been observed [31,35,40,45].

#### 3.2.8. Preoperative Differential Diagnostics and the Use of Core Needle Biopsy

Diagnosing LGASC in core needle biopsy (CNB) can be challenging, as it is frequently associated with benign sclerosing lesions, such as sclerosing papilloma, complex sclerosing lesion/radial scar, syringomatous adenoma, tubular carcinoma, and adenomyoepithelioma, complicating histopathological assessment. Additionally, many cytological criteria for malignancy are not applicable to LGASC [4,12,23,24,31,49].

Histologically, LGASC presents as a variable mixture of well-formed glands and solid cell strands with squamous differentiation and typically minimal cytologic atypia, sharing common features with radial scar and complex sclerosing lesions. The associated stroma may be fibrotic or cellular. There may also be frequent peripheral lymphoid aggregates and intratumour lymphocytic infiltrates. These similarities complicate differential diagnosis, especially in small CNB samples, where the overall features and architecture of LGASC may not be fully evaluable, leading to frequent misdiagnoses in CNB [1,24,27,29,31,46,50].

The most challenging differential diagnosis is a syringomatous tumor of the nipple, as it has a similar histological appearance and immunophenotype to LGASC. The different location is typically the key to distinguishing these lesions. However, since LGASC also tends to occur in the periareolar region, these two lesions may be confused with one another [29,51].

The identification of epithelial differentiation in metaplastic breast carcinomas requires the use of a panel of immunohistochemical markers, for example, cytokeratin and myoepithelial cell markers.

#### 3.2.9. Surgical Treatment

Due to the challenges in preoperative diagnostics, a diagnostic resection is often initially performed for LGASC. Even when LGASC is correctly diagnosed preoperatively, patients are usually candidates for breast-conserving surgery (BCS), as the tumors are predominantly small (mean size: 19 mm at diagnosis). In cases reported in the 2000s, 82% (107 of 131) of patients have undergone BCS or excision as their primary surgical treatment.

Notably, axillary procedures are often not performed alongside excision and are sometimes omitted entirely, reflecting published data indicating a very low risk of lymph node metastasis [24]. Based on the literature review, SLNB in such scenarios does not appear to be necessary, especially considering recent advancements in understanding the limited role of axillary lymph node staging in breast cancer treatment overall.

#### 3.2.10. Postoperative Histopathological Assessment

The typical appearance of LGASC is an irregular, firm lesion with a pale to tan-yellow cut surface. Microscopically, LGASC exhibits a mixture of tubular, well-formed glands and solid cell clusters with squamous differentiation in a random pattern. These are surrounded by a pale, mildly edematous stroma containing bland spindle cells, typically accompanied by a peripheral lymphocyte infiltrate [1]. The stroma can vary, appearing cellular, collagenous, or hyalinized, and the spindle cells sometimes express cytokeratins and myoepithelial markers. A variable layer of myoepithelial cells, typical of benign lesions, may be observed around the tubules. Both glandular and solid areas, as well as spindle cells, exhibit mild atypia, few mitoses, and a low Ki67 rate. Small lymphocyte clusters are common, and, rarely, areas of chondroid or osseous metaplasia may be present [7,12,25].

LGASCs are generally triple negative, lacking ER, PR, and HER2 expression, with very few exceptions [32,43]. Ki-67 is generally low, from 5% to 15%. The typical immunophenotype includes the absence of hormone receptor staining, along with the positive expression of E-cadherin, p63, and SMA and at least one of the cytokeratins CK AE1/3, CK5/6, CK7 or CK14. SMM usually appears to be negative. The staining cannot be used accurately in diagnostics, as Kawaguchi and Shin aptly point out: LGASC consistently stains in an inconsistent manner [7,24,44].

The relationship between LGASC and benign sclerosing breast lesions is particularly interesting, as they share adenosquamous proliferation (ASP) as a common feature. ASP has an almost identical histological and immunohistochemical profile to LGASC. In sclerosing lesions, ASP is considered a clonal proliferation and is thought to be a potential, though not obligatory, precursor to LGASC [24,36].

The differential diagnosis of LGASC and benign lesions can be challenging. Since LGASC is most often p63-positive, confusion with benign proliferative lesions is possible. However, in LGASC, myoepithelial staining is typically reduced, focal, or absent in invasive areas, whereas benign lesions typically exhibit strong and continuous myoepithelial staining. As ER and PR are usually positive in benign lesions and negative in LGASC, these markers can help distinguish LGASC from most benign lesions. LGASC typically shows positive staining for CK5/6, highlighting its basal-like characteristics. Benign lesions, such as sclerosing adenosis or usual ductal hyperplasia, may also stain positive for CK5/6 but are less likely to demonstrate the infiltrative pattern characteristic of LGASC [24,44].

#### 3.2.11. Adjuvant Therapy

Due to the lack of specific treatment guidelines, reported adjuvant treatments after surgical excision vary substantially. Following mastectomy, adjuvant treatment is typically not initiated. Patients undergoing BCS have often, but not always, received radiation therapy. A few patients have been reported to receive chemotherapy. Since LGASC is typically of the triple-negative subtype, antihormonal therapy is generally not beneficial. However, in rare cases of hormone receptor positivity, adjuvant hormonal therapy is usually initiated.

#### 3.2.12. Prognosis

The prognosis associated with LGASC is excellent. The only patient with reported DM and LGASC-related death was included in van Hoeven et al.’s data from 1993 [12]. Since then, 92 patients have been reported with follow-up data, with no late distal recurrences of LGASC-related deaths.

LGASC is thought to have a high propensity for local recurrence. This perception originates from the early studies by Rosen et al. and van Hoeven et al., which reported recurrence risks of 36% (4/11) and 20% (5/25), respectively [11,12]. However, if these studies—possibly involving some of the same patients—are excluded from the overall data, only three local recurrences in 92 patients (3.3%) have been reported, presenting a much more indolent picture of the disease.

Cartagena et al. report the only patient with recurrent LGASC after radical initial excision. A 37-year-old woman developed local recurrence 98 months after the initial diagnosis. Her 6 mm LGASC was treated with BCS and negative margins. No distant metastases were observed [42].

Scali et al. present a case in which a patient included in their study had previously undergone a diagnostic excision with a benign diagnosis. Six years later, the patient discovered a mass in the same area. The mass was removed and diagnosed as LGASC. At that point, the initial mass was reviewed and also confirmed to be LGASC. The margins from the first operation were not reported [26].

In two cases, the disease was reported to progress to recurrent intermediate- or high-grade malignancy. Chuthapisith et al. present a case in which a patient’s LGASC developed a local recurrence four years after the primary excision. The recurrent tumor predominantly consisted of osteosarcoma with an adenosquamous carcinoma component [25]. Sae-Kho et al. present the case of an 84-year-old woman who was treated with BCS for LGASC and developed recurrent intermediate-grade adenosquamous carcinoma one year after the initial treatment [40].

Moreover, Lewis et al. report a case of a patient who refused surgical treatment after a diagnostic biopsy for LGASC. The patient developed lymph node metastasis 18 months after the diagnosis and was reported to be alive 19 months after the metastasis was detected [44].

## 4. Discussion

Given its rarity, the current literature rarely discusses LGASC as a distinct entity, separate from other MpBCs. Consequently, its unique behavior often goes unrecognized. However, as demonstrated in this article, LGASC is clearly a separate entity with an extremely favorable prognosis, differing significantly from MpBCs in general. Its treatment should reflect its indolent nature, avoiding overtreatment.

This study also presents a single-center series of three patient cases, which vividly demonstrate the rarity of LGASC, its characteristic features, and the potential pitfalls in its treatment.

Previously, it has been estimated that 1 out of 4000 to 5000 breast cancer cases are LGASC [3,4]. The three cases presented out of 6462 breast cancer patients suggest that this estimate is reasonably accurate. However, diagnosing LGASC appears to be challenging, making it likely that some cases have not been correctly recognized or classified. As a result, the true incidence of LGASC remains somewhat uncertain. At the very least, LGASC can be regarded as a very rare subtype of breast cancer.

The clinical presentation of LGASC is typical: a small, round, palpable periareolar lump was detected in two out of three patients. In one case, the tumor was not palpable, likely due to the patient’s obesity and very large breast size, combined with the tumor’s small size (9 mm).

As described in the literature, the findings from imaging studies were nonspecific. In one patient, imaging revealed extensive microcalcifications. However, it was later determined that the microcalcifications were located outside the LGASC tumor and were associated with benign mastopathy. This finding led to an overestimation of the tumor size and ultimately resulted in a mastectomy.

Each patient underwent CNB prior to surgery. As frequently reported in the literature, CNB resulted in an incorrect initial diagnosis in all three cases.

All three cases of LGASC were primarily of the triple-negative subtype, although in two cases, hormonal receptors showed focal positivity. It should be noted that most of the current literature originates from an era when a positivity rate under 10% was considered negative. If that threshold had been used as the criterion in these cases, all three would have been classified as triple negative. Lewis et al. found focal ER and PR positivity in 5 out of the 20 examined samples [44].

Two of the three patients underwent BCS, while one with extensive microcalcifications underwent mastectomy. It is well documented in the literature that LGASC often presents alongside benign lesions with microcalcifications, as was the case here. It remains uncertain whether recognizing this earlier could have allowed BCS in this patient also.

Two of the patients underwent SNB during breast surgery, while one patient, whose diagnosis was uncertain at the time of surgery, underwent SNB as a separate procedure. None of the patients had lymph node metastases, which aligns with the literature, as only one case of axillary metastasis has been reported. The very same patient also had distant metastasis at the time of diagnosis and later succumbed to metastatic disease [12].

All three patients were discussed postoperatively in a multidisciplinary team (MDT). The discussions and decisions regarding adjuvant treatments effectively highlight the challenges of treating a rare and atypical subtype of breast cancer. Two of the three patients who underwent BCS received radiation therapy, in line with general breast cancer treatment guidelines. However, based on the current literature, the benefit of radiotherapy appears to be limited, as the risk of local recurrence seems to be significantly lower than initially presumed in the earliest studies of LGASC.

In general, adjuvant hormonal treatment is not indicated for LGASC, as it typically exhibits a triple-negative phenotype. However, two of the patients showed focal positivity for hormonal receptors on histopathological assessment, and adjuvant hormonal therapy was initiated.

Based on the literature, chemotherapy does not appear to be necessary for these patients. However, chemotherapy was recommended for two patients by the MDT. In one case, the decision was re-evaluated by the oncologist afterward, and chemotherapy was ultimately not initiated. In the other case, the indolent nature of LGASC was not recognized by the MDT, and the patient was recommended chemotherapy based on an MpBC diagnosis.

The patients have been monitored for six, seven, and ten years, respectively, and all remain disease-free. The literature on the risk of local recurrence in LGASC is controversial. The earliest published studies reported a very high risk of local recurrence, but later studies suggest that the risk is substantially lower, regardless of the adjuvant therapy the patients received.

## 5. Conclusions

Low-grade adenosquamous carcinoma (LGASC) of the breast is an indolent subtype of metaplastic breast cancer. Due to its benign behavior, it should be clearly distinguished from other types of metaplastic cancer. Unlike more common MpBC subtypes, LGASC metastasizes extremely rarely. Based on the literature, the recommended surgical treatment for LGASC is breast-conserving therapy without axillary staging, aiming for negative surgical margins. Adjuvant radiation therapy is likely not beneficial, and neither adjuvant hormonal therapy nor chemotherapy are indicated.

## Figures and Tables

**Figure 1 cancers-16-04246-f001:**
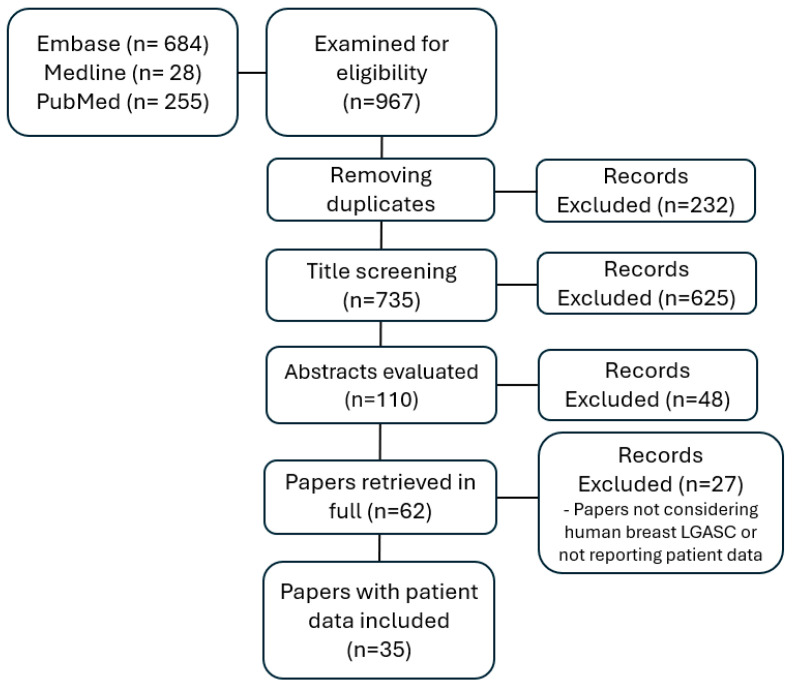
Flow chart of the systematic review. LGASC = Low-Grade Adenosquamous Carcinoma.

**Table 1 cancers-16-04246-t001:** Clinical presentation and radiographic findings.

Patient	Age	Diagnosis	Location	Size (Clinical)	Ultrasound	Mammography	CNB	MRI	Size (Final)
1	73	Mass	Periareolar	15 mm	Hypoechoic, round (16 mm)	Spiculated (20 mm)	Radial scar	no	20 mm
2	68	Mass	Periareolar	30 mm	Round, solid (30 mm)	Round, indistinct margins, microcalcifications around (30 mm)	Fibroadenoma (repeat CNB: malignancy of uncertain type)	yes *	30 mm
3	63	Screening	Lateral part of breast	not palpable	Unspecific lesion with indistinct margins	Unspecific lesion (12 mm)	Ductal carcinoma	no	7 mm

* MRI revealed microcalcifications, which were considered possible satellite tumors. MRI = magnetic resonance imaging, CNB = core-needle biopsy.

**Table 2 cancers-16-04246-t002:** Histopathological data of the patients.

Patient	ER	PR	HER2	Ki67	p63	Calponin	CkPAN	CK7	CK5/6	TILs	CD117
1	0%, focally 50%	0%	0	10%	+	+	+	+	+	+	−
2	0%, focally 10%	0%, focally 10%	0	20%	+	+	+	+	+	NR	NR
3	0%	0%	0	13%	−	−	NR	NR	+	+	NR

ER = estrogen receptor, PR = progesterone receptor, HER2 = human epidermal growth factor receptor 2, NR = not reported. TILs = tumor-infiltrating lymphocytes.

**Table 3 cancers-16-04246-t003:** The treatment and surveillance of the patients.

Patient	Surgery	Axilla	Axillary Lymph Node Status	Adjuvant RT	Hormonal Therapy	Chemotherapy	Follow-Up
1	BCS	SLNB	0/2	+	+	− *	6 y (disease free)
2	MT	SLNB	0/3	−	+	+ **	7 y (disease free)
3	BCS	SLNB	0/3	+	−	−	10 y (disease free)

* The patient was recommended chemotherapy by a multidisciplinary team, but eventually not initiated. ** following treatment guidelines for triple-negative breast cancer. BCS = breast-conserving surgery, MT = mastectomy, SLNB = sentinel node biopsy, RT = radiation therapy.

## Data Availability

The data that support the findings of this study are available upon request from the corresponding author. The data are not publicly available due to privacy or ethical restrictions.

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
