# Peer review of "Low-Grade Adenosquamous Carcinoma of the Breast: A Single-Center Retrospective Study and a Systematic Literature Review"

_cancers, 2024, doi:10.3390/cancers16244246_

Round 1
Reviewer 1 Report
Comments and Suggestions for Authors
First of all, I would like to thank you for inviting me to review the manuscript entitled “Low-Grade Adenosquamous Carcinoma of the Breast: A Single-2 Center Retrospective Study and A Systematic Literature Review”.
The title accurately reflects the case. The manuscript involves an important area of health and presents a clear and clinically useful message. The manuscript is well written in terms of clarity, style, and use of English and has a logical construction. The discussion section explains the case in the context of published information. The conclusions accurately explain the main clinical message. The references are appropriate and current.
Minor requests
1) The authors must add histological photos (H&E and immunohistochemistry) of the three reported cases as a supplementary file. This is mandatory.
2) On page 1, line 31, Please name the six subtypes of MBC.
3) Was the systematic review performed according to the PRISMA (“Preferred Reporting Items for Systematic Reviews and Meta-Analyses”) guidelines?
4) Was the search conducted through the PICO process? Please explain.
5) Please explain in Table 2 how ER can be 0% and focally 50% and 10% in patients 1 and 2, respectively. Do the same for PR for patient 2. After reading the manuscript, I believe that 0% is wrong.
6) In Table 3, what does v stand for in the follow-up? After reading the manuscript, I believe v should be replaced with a y, which means years.
Author Response
I sincerely appreciate the reviewer’s perceptive and insightful comments. Here are the point-to-point responses to each comment.
Comment 1: The authors must add histological photos (H&E and immunohistochemistry) of the three reported cases as a supplementary file. This is mandatory.
Response 1: Thank you for the comment. We have now added available photos as well as the description by the pathologist as a Supplementary File 1.
Comment 2: On page 1, line 31, Please name the six subtypes of MBC.
Response 2: The six subtypes (spindle cell carcinoma, squamous cell carcinoma, metaplastic carcinoma with heterogeneous differentiation, low‐grade adenosquamous carcinoma, low‐grade fibromatosis‐like carcinoma, and mixed metaplastic carcinoma) are now named in this section.
Comment 3: 3) Was the systematic review performed according to the PRISMA (“Preferred Reporting Items for Systematic Reviews and Meta-Analyses”) guidelines?
Response 3: The PRISMA guidelines were followed, when appropriate, considering the nature of literature, which consists mainly of case reports and short case series. Due to this, the analysis is descriptive. The relevant information is added to the matherials and methods -section of the manuscript as follows:
"The systematic review was conducted in accordance with the PRISMA (Preferred Reporting Items for Systematic Reviews and Meta-Analyses) guidelines, where applicable, considering the nature of the literature, which consists mainly of case reports and short case series."
and
"The following information was collected: Period in which the patient information was collected, number of patients, age, presentation and size of the tumour, surgery performed, information of local or distant metastasis at the time of diagnosis, adjuvant treatments and follow-up information. As a result, synthesis and descriptive analysis of the published cases are presented."
Comment 4: Was the search conducted through the PICO process? Please explain.
Response 4: I refer to the response 3. As the literature only consist of case reports and short case series, only a descriptive analysis was possible.
Comment 5: Please explain in Table 2 how ER can be 0% and focally 50% and 10% in patients 1 and 2, respectively. Do the same for PR for patient 2. After reading the manuscript, I believe that 0% is wrong.
Response 5: Thank you for this comment. According to the previous literature and the cases we present, LGASC may present as negative for ER and PR in general, but with focal positivity, with a very limited total area compared to the large negative area. In our view, such an expression (0% in general, x % focally) quite accurately represents this heterogenous presentation and is commonly used expression in the literature to describe this type of observation.
Comment 6: In Table 3, what does v stand for in the follow-up? After reading the manuscript, I believe v should be replaced with a y, which means years.
Response 6: Thank you for pointing this out, v should be indeed replaced with y. This is now corrected.
Reviewer 2 Report
Comments and Suggestions for Authors
Dear authors
thank you for this comprehensive update on this very rare tumor biology. Once published your article will fill a clinical need. There are however two remarks:
a) what is the aim of providing a statistical significance analysis of the case numbers (three of the 74 were diagnosed,...)? (page 2 line 74)
b) Figure 1 should contain more details. i.e. which database contributed how many cases for which search term.
That should be addressed prior to publication.
74
patients (0.05%, 95% CI, 0.01-0.014%) were diagnosed with LGASC.
Author Response
I thank the reviewer for this comments, that clearly improve the article. Here are the point-to-point responses to the comments:
Comment a) what is the aim of providing a statistical significance analysis of the case numbers (three of the 74 were diagnosed,...)? (page 2 line 74)
Response:
The true incidence of Low-Grade Adenosquamous Carcinoma is still uncertain, and single case reports do not provide information of this matter. Reporting the confidence interval provides additional context and helps convey the precision and reliability of the estimate, even in exploratory or descriptive statistics. It is useful information for readers who wish to assess the range within which the true value is likely to fall.
b) Figure 1 should contain more details. i.e. which database contributed how many cases for which search term.
Thank you for this comment. The Figure 1 is supplemented in the manuscript as requested. I wish the reviewer finds the supplemented flow chart to be satisfactory.
Reviewer 3 Report
Comments and Suggestions for Authors
The topic of the article is very interesting especially due to the rarity of the disease.
Methods are brief and clear.
The results' section should probably define the heading 3.1 as "hospital cases" or similar inmfor a better understanding. Besides the IHC pattern, a few lines of morphologic description of cases could be added too.
Table 3 in the results should explain "6v, 7v, 10v" (follow up) in the legend.
The results of Literature review (3.2) are very well structured in a logical manner, describing every step of a patient's journey.
It's been a pleasure to read.
The discussion covers all the data presented in the body of the article, and the conclusions are done with precision.
Supplementary material was well appreciated.
I think it is of a great importance to bring up more systemic literature reviews on a rare tumor subtypes.
Author Response
I thank the reviewer for comments, which clearly aim to improve the readability of the article.
Comment 1: The results' section should probably define the heading 3.1 as "hospital cases" or similar inmfor a better understanding.
Response 1: The reading is now rephrased as "Low-Grade Adenosquamous Carcinoma Patient Cases in Single Center Data". I hope this more clearly reflects the content of the section and meets the reviewer's approval.
Comment 2: Besides the IHC pattern, a few lines of morphologic description of cases could be added too.
Response 2: Thank you for this comment. Considering this and comment and the ones provided by reviewer 1, we have now added histopathological photos and descriptions in full as a Supplementary File 1. We hope the reviewer is satisfied with the presentation.
Comment 3. Table 3 in the results should explain "6v, 7v, 10v" (follow up) in the legend.
Response 3: Thank you for notifying me of this unfortunate error. This should be 6y, 7y and 10y (years) which is now corrected to the article.
Comment 4: The results of Literature review (3.2) are very well structured in a logical manner, describing every step of a patient's journey.
It's been a pleasure to read.
The discussion covers all the data presented in the body of the article, and the conclusions are done with precision.
Supplementary material was well appreciated.
I think it is of a great importance to bring up more systemic literature reviews on a rare tumor subtypes.
Response 4: We thank the reviewer warmly from these delightful comments.
Reviewer 4 Report
Comments and Suggestions for Authors
The author presents a single-center retrospective analysis of LGASC patients along with a detailed review of the literature.
The Manuscript is well written and the presented topic is potentially interesting. Before acceptance i have a few suggestions to improve the overall quality
1) Results: patient data and case presentations should be more clearly written in a separate section (Tables are not sufficient to show the results)
2) Histopathological description of LGASC lacks a differential diagnosis with other benign conditions of the breast. Since this histotype is by definition p63 positive (p63 is a squamous epithelium marker but also a marker of myoepithelial cells) it could be mistaken for a benign proliferation (adenosis...). Therefore, ER and PGr receptor are also useful in this diagnosis, since their negativity helps to suspect a triple-negative phenotype. Please discuss in deep these differential diagnostic considerations.
Author Response
We thank the reviewer for these comments, which clearly aim to improve the presentation of our article. Here are our point-to-point responses to the Reviewers comments.
Comment 1: Results: patient data and case presentations should be more clearly written in a separate section (Tables are not sufficient to show the results)
Response 1: We thank the reviewer for this comment. However, the decision to use tables for presenting the results was explicitly mandated by the journal, rather than the authors' choice, so unfortunately we are unable to implement this suggestion. However, we have added histopathological photos and full descriptions as a Supplementary File, and we sincerely hope that this solution will sufficiently satisfy the reviewer.
Comment 2: Histopathological description of LGASC lacks a differential diagnosis with other benign conditions of the breast. Since this histotype is by definition p63 positive (p63 is a squamous epithelium marker but also a marker of myoepithelial cells) it could be mistaken for a benign proliferation (adenosis...). Therefore, ER and PGr receptor are also useful in this diagnosis, since their negativity helps to suspect a triple-negative phenotype. Please discuss in deep these differential diagnostic considerations.
Response 2: We thank the reviewer for bringing up this important matter. We added the following text to discuss this important matter in more detail:
The differential diagnosis between LGASC and benign lesions can be challenging. Since LGASC is most often p63-positive, confusion with benign proliferative lesions is possible. However, in LGASC, myoepithelial staining is typically reduced, focal, or absent in invasive areas, whereas benign lesions typically exhibit strong and continuous myoepithelial staining. As ER and PR are usually positive in benign lesions and negative in LGASC, these markers can help distinguish LGASC from most benign lesions. LGASC typically shows positive staining for CK5/6, highlighting its basal-like characteristics. Benign lesions, such as sclerosing adenosis or usual ductal hyperplasia, may also stain positive for CK5/6 but are less likely to demonstrate the infiltrative pattern characteristic of LGASC.
Reviewer 5 Report
Comments and Suggestions for Authors
The subject of the submitted article, diagnosis and treatment of LGASC, is definitely an important one and although the publication of the three cases the authors are presenting, is of limited value, the literature search the authors have done makes the manuscript interesting. In an entity with an incidence of 0.05% of breast cancer cases this article can become a reference for the reader who has a case in clinical routine. And the message this manuscript is supposed to transport case to be supported: LGASC does not need radical local and/or systemic treatment. A few issue have to be addressed before the manuscript can be recommended for publication.
The use of the abbreviation MBC for metaplastic breast cancer is irritating, because it is usually used for metastatic breast cancer. I would suggest not to use an abbreviation for the term metaplastic breast cancer.
In the introduction the authors state that there are 6 subtypes of metaplastic breast cancer and that all are aggressive except LGASC. Please elaborate more on that and name the 5 other subtypes. If you think that the introduction becomes too long, maybe shift this topic to the discussion.
The authors bare stating that there is no guideline for LGASC. That is unfortunately not correct. In the NCCN guidelines for breast cancer, version 2.2024, LGASC is specifically addressed and the need for adjuvant therapy is questioned. Please add this information for the reader.
Editorial: In table three in the column Follow Up the abbreviation after what is supposed to be years is"v". Should that not be a "y"? And although I am myself not a native speaker the adverb "mammographically" sound wrong. Maybe that can be checked by the editorial office.
Author Response
We thank the reviewer for insightful comments. Here are our point-to-point responses to each comment.
Comment 1. The use of the abbreviation MBC for metaplastic breast cancer is irritating, because it is usually used for metastatic breast cancer. I would suggest not to use an abbreviation for the term metaplastic breast cancer.
Response 1: We understand the confusion with more commonly used meaning of the abbreviation. Since the term metaplastic breast cancer is repeatedly used in the article, we suggest using "MpBC" instead of "MBC" to maintain sufficient readability, and wish that the reviewer is satisfied with this alternative abbreviation.
Comment 2: In the introduction the authors state that there are 6 subtypes of metaplastic breast cancer and that all are aggressive except LGASC. Please elaborate more on that and name the 5 other subtypes. If you think that the introduction becomes too long, maybe shift this topic to the discussion.
Response 2: Thank you for this comment. We added the names of the 5 other subtypes (spindle cell carcinoma, squamous cell carcinoma, metaplastic carcinoma with heterogeneous mesenchymal differentiation, low‐grade adenosquamous carcinoma, low‐grade fibromatosis‐like metaplastic carcinoma, and mixed metaplastic carcinoma) to the introduction, and emphasized the difference in natural behaviour between LGASC and other subtypes of MpBC as follows:
"LGASC is considered an indolent type of breast cancer, in contrast to the typically aggressive nature of other subtypes of MpBCs "
Comment 3: The authors bare stating that there is no guideline for LGASC. That is unfortunately not correct. In the NCCN guidelines for breast cancer, version 2.2024, LGASC is specifically addressed and the need for adjuvant therapy is questioned. Please add this information for the reader.
Response 3: We thank the reviewer for pointing this out. If we understand correctly, the NCCN guideline does not address LGASC as a separate entity but mentions it once alongside other breast cancer types with a favorable prognosis, stating that systemic treatments may not be necessary for localized disease, although the data on this is limited, and the need for radiation therapy is not addressed at all. Thus, we believe that at least the phrasing 'no clear guideline' should be considered justified.
However, as the reviewer correctly state, it is appropriate to add this information to the article, and we wrote as follows:
NCCN breast cancer guidelines state that there is limited available data on the treatment of LGASC, but that it has a favourable prognosis even without systemic adjuvant therapy. The guideline does not address LGASC as a separate entity but mentions it alongside other breast cancer types with a favorable prognosis, stating that systemic treatments may not be necessary for localized disease, although the data on this is limited, and need for radiation therapy is not addressed.
Comment 4: In table three in the column Follow Up the abbreviation after what is supposed to be years is"v". Should that not be a "y"?
Response 4: Thank you, the reviewer is absolutely right. The unfortunate error is now corrected.
Comment 5: And although I am myself not a native speaker the adverb "mammographically" sound wrong
Response 5: We believe the term 'mammographically' is commonly used in English; however, if the editorial team finds it unsuitable for this purpose, we are happy to replace it with an alternative term they suggest.